# Quantification of basal ice microbial cell delivery to the glacier margin

Mario Toubes-Rodrigo<sup>1,2</sup>, Simon J. Cook<sup>1,2</sup>, David Elliott<sup>3</sup>, Robin Sen<sup>1</sup>

<sup>1</sup>School of Science and the Environment, Manchester Metropolitan University, Manchester, M1 5GD, United Kingdom <sup>2</sup>Cryosphere Research at Manchester, Oxford Road, Manchester, United Kingdom

<sup>3</sup>Environmental Sustainability Research Centre, University of Derby, DE22 1GB, United Kingdom

Correspondence to: Mario Toubes-Rodrigo (m.rodrigo@mmu.ac.uk)

## Abstract

5

We present the first assessment of microbial cell discharge from sediment-laden glacier basal ice. At Svínafellsjökull, a temperate valley glacier in Iceland, approximately 10<sup>17</sup> cells a<sup>-1</sup> are transferred through basal ice to the proglacial environment,

and between  $10^1$  and  $10^6$  cells g<sup>-1</sup> basal ice were cultured from our samples under laboratory conditions. We suggest that the delivery of viable cells and dead microbial matter to proglacial ecosystems could be playing a crucial role in soil formation and primary succession during deglaciation, but further quantification of cell transfer from a range of glacier contexts is required.

#### 1. Introduction

- Glaciers and ice sheets are important ecosystems that support microbial life (Hodson et al., 2015). Although most work hitherto has focused on life at the glacier surface, the subglacial environment represents a potentially important but poorly understood microbial niche (Sharp et al., 1999). Debris-laden glacier basal ice plays an important role in sediment transfer to the ice margin because it is in contact with the glacier substrate and is involved in most of the geomorphological work achieved by a glacier (Knight, 1997). Whilst there is a body of research on the delivery of inorganic materials (i.e. bedrock and sediment) to the ice-
- marginal environment, and the associated development of landforms and sediments (Cook et al. 2011a), there is a dearth of information on the delivery of organic material, including microbes, to the glacier margin. An intriguing possibility is that microbes released from glaciers could play an important role in proglacial soil and vegetation development, yet despite a wealth of studies on vegetation succession in deglaciating environments (Brown & Jumpponen 2014; Shivaji et al. 2011), this has not yet been fully explored. Importantly, basal ice melt-out could deliver viable microbiota to the ice margin that serve as
- inoculum, potentially accelerating pedogenesis as glaciers recede (Kaštovská et al., 2007). We report the first quantification of microbial discharge to a glacier margin, and demonstrate that there is viable microbial inoculum released to the proglacial environment. Results from Montross et al. (2014) imply that microbial numbers should scale with debris content in the basal ice, thus predicting that rate of cell discharge to the ice margin will be positively correlated with debris content. Few studies have quantified sediment discharge from basal ice due to incomplete knowledge of subglacial
- systems (Wainwright et al., 2015), although this has been achieved at a few well-studied sites (e.g. Cook et al. 2010, 2011b). Likewise, there is limited literature on quantified microbial content, either viable or total, within basal ice (Lawson et al., 2015;

Montross et al., 2014). To our knowledge, no studies have yet addressed both shortcomings. Our aims were to determine cell discharge from debris-bearing basal ice, and confirm that viable microbial inoculum are transferred between glaciers and proglacial ecosystems. This study was undertaken at Svínafellsjökull, Iceland, which is one of only a few well described glaciers with regards to the physical characteristics of basal ice and its formation and sediment discharge to the ice margin from individual basal ice facies (Cook et al. 2007, 2010, 2011a,b).

5

#### 2. Study site and methods

## 2.1 Svínafellsjökull

Svínafellsjökull (63°59' N, 16°52' W) is a temperate outlet glacier of the Oræfajökull ice cap, southeast Iceland (Figure 1). It descends from the ice cap summit down an icefall and terminates in a subglacial overdeepening. Historical records indicate
that between 1350 and 1500, the terminus was approximately 3.5 km further up-valley than it is today, with part of the area currently covered by the glacier used for agriculture (Ives, 2007). It is likely, therefore, that Svínafellsjökull has overridden ancient soil and vegetation, which may now constitute part of the subglacial debris load.

## 2.2 Basal ice samples

Basal ice is typically heterogeneous in physico-chemical composition (Hubbard et al., 2009; Knight, 1997) and could thus be

- hypothesised to exhibit spatio-temporal variation in numbers and diversity. Unfortunately, previous studies of subglacial microbiology have not always provided adequate descriptions of the ice and sediment being sampled. Basal ice at Svínafellsjökull has been described in detail by Cook et al. (2007, 2010, 2011b) and, broadly, comprises two main facies. Firstly, 'stratified facies' that is characterised by high sediment content (approximately 32 % by volume) displays a distinctive layered appearance (Figure 1b) and is localised as layer up to 4 m thick in contact with the glacier bed around the southernmost
- part of the glacier margin. Cook et al. (2007, 2010) demonstrated that much of this facies was formed through glaciohydraulic supercooling in the terminal overdeepening. Cook et al. (2007, 2010) also described a second population of stratified facies formed by regelation, but this was not observed in the present study. Secondly, 'dispersed facies' (Figure 1c; Cook et al., 2011b) is characterised by lower sediment content (1.6 % by volume), but is found pervasively around the glacier margin in a layer up to 25 m thick, either in contact with the glacier bed, or overlying the stratified facies. Cook et al. (2011b) suggested
- that dispersed facies was formed by tectonic entrainment of sediment at the base of the icefall, followed by regelation, deformation and strain-related metamorphism.

Three stratified and three dispersed facies samples were collected aseptically from Svínafellsjökull in April 2015 following methods outlined in Toubes-Rodrigo et al. (2016) (sample locations on Figure 1a). The distribution, type and thickness of basal ice was similar to that described by Cook et al. (2007, 2010, 2011b). Ice samples were allowed to melt over 48 hours at 4°C

and sediment-bearing water was poured into 50 ml sterile Falcon tubes. To preserve the cells, formamide was added to a final 4% v/v concentration. Samples were kept at 4°C until further analysis.

5

# 2.2 Cell counts

Samples were allowed to melt at 4°C over 48 h and fixed in formaldehide at a final concentration of 4% v/v and stored at 4°C until further analysis. Fixed samples were sonicated for 3 minutes using an S-Series Table Top ultrasonicator (Sonicor Inc. USA) at maximum intensity to release the cells from the sediment grains, and 1 ml was extracted from the supernatant and filtered using 0.2 µm Whatman® Nuclepore<sup>TM</sup> Track-Etched Membranes. For cell enumeration in sediment entrained in basal ice samples, dried filters were stained using 4'-6-diamidino-2- phenylidole (DAPI) (1mg ml<sup>-1</sup>) and covered with Leica type P

ice samples, dried filters were stained using 4'-6-diamidino-2- phenylidole (DAPI) (1mg ml<sup>-1</sup>) and covered with Leica type P immersion liquid prior to examination in an epifluorescence microscope Nikon Eclipse E600 (Nikon, Japan) (Lunau et al., 2005)

## 2.3 Colony forming units (CFU) count in sediment

In order to analyse if the community inhabiting the basal ice was alive and viable, a 1g sub-sample was diluted in 9 ml of saline buffer (0.85 % w/v NaCl) and vortexed for 2 minutes at full speed (Cliffton Cyclone MIX5050 vortexer, SLS, UK). Serial dilutions were performed in the saline buffer from 10<sup>-1</sup> to 10<sup>-5</sup>, and transferred for isolation onto 1:10 Tryptone Soy Agar (TSA) plates with cycloheximide (100 mg/ml) (Elliott et al., 2015). Plates were incubated at 4°C for 5 weeks and colonies were counted.

# 15 2.4 Cell discharge calculations

Svínafellsjökull is one of the few glaciers where basal ice sediment transfer has been quantified. Cook et al. (2010) calculated that sediment flux through the stratified facies was 4.8 to 9.6 m<sup>3</sup> m<sup>-1</sup> a<sup>-1</sup> (i.e. the volume of sediment transferred per metre of basal ice exposure around the glacier margin, per year), with the range in values determined by the range in previously measured glacier velocities. The equivalent value for sediment discharge was calculated as 207 to 414 m<sup>3</sup> a<sup>-1</sup> (Cook et al.,

- 2011b). These values relate specifically to the stratified basal ice formed by glaciohydraulic supercooling. For the current study, stratified basal ice associated with regelation was not observed, as it was by Cook et al. (2010), and so is not considered here. Cook et al. (2010, 2011b) calculated a sediment flux of 1.0 to 2.0 m<sup>3</sup> m<sup>-1</sup> a<sup>-1</sup>, and a sediment discharge of 42 to 84 m<sup>3</sup> a<sup>-1</sup>, for this so-called 'non-supercool' basal ice, so its contribution to cell discharge would have been much lower even if it were present. Cook et al. (2011b) calculated that sediment flux through the dispersed facies was 0.81 to 1.62 m<sup>3</sup> m<sup>-1</sup> a<sup>-1</sup>, and that
- sediment discharge was 1635 to 3270 m<sup>3</sup> a<sup>-1</sup>. The very high sediment discharge relative to stratified facies is because dispersed ice is found ubiquitously around the glacier margin in metres-thick sequences, despite its low sediment content. We follow glacial sediment transfer convention by quantifying analogous cell transfer values as 'cell discharge', in cells m<sup>-1</sup> a<sup>-1</sup>, and 'cell flux', in cells m<sup>-1</sup> a<sup>-1</sup>. To calculate these, cell concentration, expressed as the number of cells per kilogram of sediment (cells kg<sup>-1</sup>), was multiplied by the sediment discharge (m<sup>3</sup> a<sup>-1</sup>) or flux (m<sup>3</sup> m<sup>-1</sup> a<sup>-1</sup>), and then multiplied by sediment
- density (kg m<sup>-3</sup>). The geology of the study region is dominated by basalt ( $\rho_{\text{basalt}} = 2800-3000$  kg m<sup>-3</sup>) and hyaloclastite

 $(\rho_{hyaloclastite} = 2750 \text{ kg m}^{-3})$  (Magnússon et al., 2012). For the purpose of this work, we assume a sediment density ( $\rho_{sediment}$ ) of 2800 kg m<sup>-3</sup>.

## 3. Results

# 3.1 Cell discharge from basal ice

5 Cell counts reveal that stratified facies contains  $1.3 \times 10^7$  cells g<sup>-1</sup> of sediment (n = 3;  $\sigma$  = 2.7 x 10<sup>6</sup>), whilst dispersed facies contains  $1.4 \times 10^7$  cells g<sup>-1</sup> of sediment (n = 3;  $\sigma$  = 5.7 x 10<sup>6</sup>). Hence, stratified and dispersed facies have similar cell counts and no statistically significant differences were observed (t-test p-value = 0.72). Cell counts in supernatant (data not shown) were insignificant compared with sediment.

Cell discharge associated with the sediment within stratified facies is between 7.4 x  $10^{15}$  to 1.5 x  $10^{16}$  cells a<sup>-1</sup>, and is between

10  $6.6 \times 10^{16}$  to  $1.3 \times 10^{17}$  cells a<sup>-1</sup> for dispersed facies. Hence, 10.1% of cell discharge per year is associated with stratified facies, and 89.9% is associated with dispersed facies. The cell flux through stratified facies is between  $1.8 \times 10^{14}$  and  $3.5 \times 10^{14}$  cells m<sup>-1</sup> a<sup>-1</sup>, and for dispersed facies this value is between  $3.3 \times 10^{13}$  and  $6.6 \times 10^{13}$  cells m<sup>-1</sup> a<sup>-1</sup>. Hence, where stratified facies is present, cell flux is 5.4 times more cells than dispersed facies, but because of a more limited distribution around the ice margin, overall cell discharge is much lower than dispersed ice.

#### 15 3.2 Enumeration of viable cells

For both stratified and dispersed facies, CFU values were similar without statistically significant differences between them (t-test p-value = 0.43) and overlap can be observed in the box and whisker plot in Figure 2. Nevertheless, dispersed basal ice counts were more variable (n = 3,  $\bar{x}_{dispersed} = 5.9 \times 10^4$ ,  $\sigma^2_{dispersed} = 1.2 \times 10^{10}$ ) with three orders of magnitude difference between samples D1 and D3, whereas counts from stratified facies (n = 3,  $\bar{x}_{stratified} = 5.9 \times 10^4$ ,  $\sigma^2_{stratified} = 7.9 \times 10^9$ ) were more consistent, with less than an order of magnitude of difference between samples.

4. Discussion and conclusions

We have demonstrated that there are large numbers of cells ( $\bar{x}_{stratified} = 1.3 \times 10^7$  cells g<sup>-1</sup>,  $\bar{x}_{dispersed} = 1.4 \times 10^7$  cells g<sup>-1</sup>) associated with sediment trapped in basal ice at Svínafellsjökull, both for the stratified and dispersed facies. For comparison, Lawson et al. (2015) quantified cell counts in the basal ice of four different glaciers comprising examples of polythermal, warm- and

25

20

cold-based glaciers, and found lower cell counts, on the order of  $10^5$  cells g<sup>-1</sup>; the highest values were found in the warm-based glacier, Engabreen, whereas the lowest values were found in the cold-based Joyce Glacier. One possible explanation for the high cell counts at Svínafellsjökull is that this glacier likely overrode soils and vegetation, and associated microbiota, during its Little Ice Age advance (Ives, 2007). Nonetheless, cell numbers within basal ice are much lower than in proglacial soil, where typical cell numbers would be on the order of  $10^8$  cells g<sup>-1</sup> (Shivaji et al., 2011).

Successful cultivation of microorganisms in 1:10 TSA plates at 4°C demonstrates that microbiota present within basal ice are viable. These microorganisms, once released to the ice-marginal environment, would be able to proliferate and assist in the initiation of soil formation as the glacier recedes (Brown and Jumpponen, 2014; Rime et al., 2016). It is noteworthy that only between 1-3% of the total microbiota can be isolated using traditional methods (Armougom and Raoult, 2009), so although only up to 10<sup>5</sup> bacterial cells per gram of sediment were observed to have grown, the actual number of viable cells is likely to

- be up to 100-fold higher. Hence, our results represent conservative estimates of the delivery of viable cells to the ice margin. For the first time, we have quantified microbial cell transfer to a glacier margin through the basal ice layer. For stratified facies, the cell flux is between  $1.8 \times 10^{14}$  and  $3.5 \times 10^{14}$  cells m<sup>-1</sup> a<sup>-1</sup>, which corresponds to a total discharge of  $7.4 \times 10^{15}$  to  $1.5 \times 10^{16}$  cells a<sup>-1</sup>. For dispersed facies, the cell flux is between  $3.3 \times 10^{13}$  and  $6.6 \times 10^{13}$  cells m<sup>-1</sup> a<sup>-1</sup>, which corresponds to a cell
- discharge of 6.6 x 10<sup>16</sup> to 1.3 x 10<sup>17</sup> cells a<sup>-1</sup>. The cell flux is one order of magnitude higher in stratified facies than in dispersed facies, which results from the higher sediment content in stratified ice, in agreement with the findings of Montross et al. (2014) who also found that cell counts increased with sediment content. However, the overall cell discharge from stratified ice is much lower due to the far greater thickness and abundance of dispersed facies around the glacier margin. It is clear that different ice facies deliver different amounts of cells to the glacier margin, indicating that future studies of basal ice microbiology should
- account for ice facies characteristics, as has been the norm for glaciological studies of basal ice (e.g. Hubbard et al., 2009). The subglacial environment delivers large amounts of dead and viable cells to the ice margin at Svínafellsjökull. Dead cells represent a potentially important nutrient resource to the proglacial environment, which may promote colonisation by pioneering communities. Our results represent an individual case study based on one temperate Icelandic glacier. As demonstrated by Lawson et al. (2015), cell numbers in basal ice vary greatly according to glacier thermal regime, and other
- factors, such as substrate type (e.g. sediment vs. bedrock) and lithology, may also play important roles in determining the nature and size of subglacial microbial communities. Even within our dispersed facies samples, there is variability in CFU counts between samples (Figure 2). Hence, we recommend that similar studies be performed at other sites with different glaciological characteristics to gain a better appreciation of cell transfer to the margins of glaciers.

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

# 7613(1999)027<0107:wbpagb>2.3.co;2, 1999.

Shivaji, S., Pratibha, M. S., Sailaja, B., Hara Kishore, K., Singh, A. K., Begum, Z., Anarasi, U., Prabagaran, S. R., Reddy, G. S. N. and Srinivas, T. N. R.: Bacterial diversity of soil in the vicinity of Pindari glacier, Himalayan mountain ranges, India, using culturable bacteria and soil 16S rRNA gene clones, Extremophiles, 15(1), 1–22, doi:10.1007/s00792-010-0333-4, 2011.

5 Toubes-Rodrigo, M., Cook, S. J., Elliott, D. and Sen, R.: 3.4. 1. Sampling and describing glacier ice, in Geomorphological tehcniques, edited by J. . Cook, S.J., Clarke, L.E. Nield, British Society for Geomorphology, London., 2016. Wainwright, J., Parsons, A. J., Cooper, J. R., Gao, P., Gillies, J. A., Mao, L., Orford, J. D. and Knight, P. G.: The concept of transport capacity in geomorphology, Rev. Geophys., 53(4), 1155–1202, 2015.