# Peer review of "Quantification of basal ice microbial cell delivery to the glacier margin"

_Biogeosciences, 2016_

## Referee Comment (RC1) · Anonymous Referee #1 · 11 Nov 2016

Toubes-Rodrigo et al present evidence in support of their assertions regarding the delivery of cells from basal ice to the glacier forefield of an Icelandic glacier. The manuscript is clearly structured, and is to be commended for integrating key glaciological constraints on subglacial microbiology. Unfortunately I have a number of major reservations about the manuscript in its present form. Principally these are:

1. Claims of cell discharge relate to total cells derived from DAPI counts and viable cells from CFU counts obtained from the inoculation of a specific growth medium under one set of incubation conditions for five weeks. While total counts from microscopy are acceptable, I firmly disagree with the notion that the authors have determined viable counts, and whether the procedures employed are adequate to answer their experimental question on the following grounds:

[Figure]

A: Viable does not mean culturable. Consider the very paradox of the acronym "Viable But Not Culturable" which has been explored for >30 years by many investigators. As the issue of VBNC sets out, one of the challenges of contemporary microbial ecology is understanding the gap between who appears on your agar plate (culturable), who is present (total, including dead cells) and who might live in situ (viable) and those who are able to live in situ but not on your agar plate (VBNC).

The paper needs to take into account that viability is non synonymous with culturability. Here culturability under one set of conditions is presented. This means very little for quantitative estimation of viability. If one is minded to determine the abundance of viable cells within an environmental habitat, very different tools are required, typically in the vein of Live/Dead stains and microscopy. These are not without their problems of course.

B: One set of conditions are tested: 10% TSA, 4 deg C for five weeks. No data is presented setting out whether this set of conditions is representative or optimal. What assurance does the reader have that this protocol provides consistent counts?

C: So if viability itself is not quantified what about culturability itself - is it meaningful? What does growth in vitro really tell us about those cells' ability to colonize proglacial environs? I believe it was the eminent microbiologist John Postgate who stated that "every colony is an artefact". It is difficult to convincingly aruge that culture of cells in vitro under one set of fixed conditions necessarily provides quantitative insights to the in situ actuality.

D: No information is provided on the community composition of the inhabitants of the basal ice or the proglacial habitats they may be discharged into. Clearly, not all microbes have the same potential to colonize an environment. The ecological impact of inoculating a trillion cells which cannot persist and grow in the forefield will be very different to just one cell immured in basal ice which can also thrive in the forefield. As such, the implications of mass transfer of cells are poorly developed, and assume

equivalency of outcome across what are very different scenarios: are these cells likely to pioneer the forefield community's development, or are they simply a source of nutrients as necromass? Or will dormant cells provide a long term repositry of genetic potential for later stages of soil development? Very different ecological scenarios arising from the physiological state and colonization potential of the source microbiota which are beyond the scope of the analyses performed. As such I feel the development of the rationale of this underlying motivation of the paper is limited in its grasp, and the paper would really benefit from careful consideration of the processes underlying the assembly of microbial communities.

2. At the heart of this paper are total counts and CFU counts. While their use coupled with expert interpretation of the basal ice facies is important, this seems a little preliminary and the conclusions drawn risk superficiality as a result. The paper would be greatly strengthened as an offering to the literature if it described the taxonomic composition of the cultured and total community. As noted above, simply dumping cells into an environment has radically different outcomes dependent on the identity of the cells.

3. How representative is this site of other locations? I appreciate that its history of circumspect glaciological investigation lends itself for this study, but considering its history of advance over soils, what lessons can be learned from this site that would be applicable to sites with very different histories?

4. L8: "We present the first assessment of microbial cell discharge from sediment-laden glacier basal ice." L28: "We report the first quantification of microbial discharge to a glacier margin, and demonstrate that there is viable microbial inoculum released to the proglacial environment"

Respectfully, I disagree with the assertion of priority made for this claim, and the emphasis provided by placing it at the start of the abstract. Starting from the seminal paper of Sharp et al (1999) microbial prevalence in basal ice has been widely documented as has its potential for inoculating forefields as well as demonstratable culturable bacteria

using a range of methodologies, as well as culture-independent strategies (e.g. Kaš-tovská et al 2007; Yde et al 2010 Ann Glaciol; Montross et al 2015 Geomic J; Rime et al 2016 ISMEJ). I would highly recommend a more circumspect statement regarding the motivation of this study which clearly and fairly asserts the scientific novelty of the work. Perhaps the emphasis of integration with basal ice extent is required?

5. L16: The authors emphasize the heterogeneity inherent to these ice facies. The methods section does not set out how the samples collected were distributed across the ice facies to describe this hetereogeneity and minimise potential biases. In short, what was the specific survey design, and the extent of replication. Fig1a goes some way to explain the number of sites sampled, but more clarity is needed here, especially on potential intra-site variation.

6. Uncertainties in sediment transfer rates. These seem pretty broad, and incur a two-fold variation in the potential discharge of cells. Can the authors justify the insights afforded by this calculation considering this considerable uncertainty?

7. How does basal ice microbial discharge scale up relative to fluxes from meltwater or till? Context could be provided here.

8. Discussion needs to draw out the insights into the ecological processes affected by cell discharge from basal ice. What does it all mean for the downstream habitat?

In summary, many of the assumptions made in this paper merit careful contemplation and the datasets presented could be supplemented by orthogonal information regarding community composition. In critiquing the work offered I really do not wish to deflate the very commendable initiative shown by the early career reesearcher in the process of learning how to correspond his work. I would hope all authors to work carefully with him to strengthen a future embodiment of this paper.

Minor comments.

L27: More detail is needed here on sampling protocol and precautions to allow readers

without access to the cited source to evaluate the protcol applied. L29: Ballpark figure provided by Shivaji et al (2011). Microbial abundance changes considerably as soil develops over a chronosequence. Perhaps your basal ice abundances matter more when meeting the depauperate bare till of the immediate glacier margin. L30: Formamide? or formaldehyde? The reader needs to be reassured the microbial population is adequately fixed for enumeration. L32: Formaldehyde L36: Using DAPI on small and dormant cell populations. What assurances does the reader have about the sensitivity of DAPI in this context given its lower quantum yield of fluorescence relative to SYBR stains?

---

## Referee Comment (RC2) · Anonymous Referee #2 · 28 Nov 2016

This paper presents total cell counts from six basal ice samples collected at the front of Svinafellsjökull, Iceland. The cell counts are then combined with previously published debris flux estimates to quantify the microbial cell flux from the basal ice layer. Compared to the supraglacial environment, there is still limited knowledge on subglacial microbial ecosystems and processes, and this paper adds new knowledge to this topic. However, in my opinion there are several issues that need to be addressed before the paper is ready for publication. I agree with Anonymous Referee #1 with regards to viable cell counts, so I will not use time to repeat this here.

General comments:

1. The first thing that attracts my attention is that the paper is unnecessarily short for a research paper in Biogeosciences. It gives the impression that the paper was intended

for another journal but somehow ended up as a submission to Biogeosciences. As the paper addresses a very specific topic within subglacial microbiology, namely total cell counts and cell flux in basal ice, I strongly recommend that the paper is expanded to provide the readers with an up-to-date overview of the current knowledge of cell abundance in basal ice and place the new cell counts and fluxes in this context. This will undoubtedly increase the impact of the paper.

2. The rationale for the study is that "basal ice melt-out could deliver viable microbiota to the ice margin that serve as inoculum, potentially accelerating pedogenesis as glaciers recede" (1,24-25). This may be true for some glaciers, but when I look at the location map (Figure 1a) it seems that the entire front of Svinafellsjökull is in contact with either a glacial river or ice-marginal lakes. Hence, the cells that melt out at the six sampling sites will most likely be washed into the glacial river and transported to a downstream sandur or into the sea. It is unlikely that they will accelerate pedogenesis as Svinafellsjökull recedes. As this is a case study, the scientific rationale should reflect what is relevant for the environment at Svinafellsjökull. I suggest that the authors put more emphasis into presenting the ice-marginal environment and a rationale that addresses the conditions at Svinafellsjökull. This will also make the paper more interesting for potential future studies on the microbial community structures in the supraglacial environment, the basal ice, the proglacial river/lakes, and the proglacial foreland at Svinafellsjökull.

3. The Introduction section is basically written as "there is a lot of knowledge about this, but little knowledge about that". This form is not very interesting to read and it seems a bit dubious at times. For instance, the authors write that "few studies have quantified sediment discharge from basal ice . . . (Wainwright et al., 2015)" (1,29-20), whereas Wainwright et al. (2015), in fact, write that "several studies [e.g., Hunter et al, 1996; Knight et al., 2002] have measured actual debris flux through the basal layer" (see page 1182 in Wainwright et al., 2015). I recommend that the authors change the form and include many more relevant references and use them in an active way (e.g.,

"Cook et al. (2010) found that . . ."). A full literature overview of cell counts conducted on different basal ice facies would be relevant either in the Introduction section or the Discussion section (maybe as a table).

4. It is stressed out throughout the paper that this is the 'first' quantification of cell flux from basal ice. If it is so important to provide the first quantification of cell flux from basal ice, the authors could just have combined the debris flux in basal ice provided by Knight et al. (2002) with the total cell count in basal ice provided by Yde et al. (2010) to produce an estimate for the basal ice delivery of cells at Russell Glacier in Greenland. This will have saved them all the fieldwork. Although it may be true that this first to make this estimation, I will suggest that the mentioning about being the 'first' paper to quantify basal ice cell flux is toned down and replaced by quantitative and qualitative comparisons between the results from Svinafellsjökull and estimates of debris fluxes and cell counts from other glaciers.

Minor comments:

1,9 and 1,10: These numbers of cell flux and cell abundance in the Abstract should be similar to numbers found in the Results section.

1,17; 1,19; 1,20; 1,30; 1,31-2,1 and other places: Include more references to support these statements.

1,20 and other places: Insert comma after "et al."

1,20-21: ". . . there is a dearth of information on the delivery of organic material, including microbes, to the glacier margin". I disagree in this statement, as there are several studies on DOC in glacial rivers.

1,25: A second paragraph should provide a literature overview on microbial abundance in glacier ice, including basal ice (see e.g. Irvine-Fynn and Edwards, 2014). There is no need to go into details about basal ice microbial diversity, expect for where the microbial diversity is relevant for culturing of cells. A third paragraph could be on debris

fluxes from basal ice.

2,1-3: "Our aims were to . . . and confirm that viable microbial inoculum are transferred between glaciers and proglacial ecosystems". It is well known that viable cells are transported by subglacial river to the proglacial environment, so it must be specified that the aim of the paper is focusing on basal ice transport of microbes to a fluvial proglacial ecosystem.

2,7-12: The site description must provide more relevant information, especially regarding the proglacial river and ice-marginal lakes. What is the meteorological regime? What is the distance of glacial retreat since the Little Ice Age? What is the contemporary average frontal retreat rate per year? What is the river discharge and suspended sediment load? Is anything known about the supraglacial or proglacial microbial communities?

2,9: The period mark is red.

2,18 and 2,23: With regards to debris content, it is more correct to use "by mass" instead of "by volume" and to present supplementary information on grain size distributions. This will also be consisted with the use of mass in the calculations (3,28-29). If the stratified or dispersed facies contain gravel, stones or boulders, it should also be noted.

2,19: Insert "a" before "layer".

2,25: How far up-glacier is the icefall?

2,26: I don't understand the term "strain-related metamorphophism". Isn't all change of ice crystals in solid-state (i.e. metamorphosis) strain-related?

2,27-28: In my opinion, the most obvious sampling strategy would have been to select and survey one to three basal ice profiles perpendicular to the basal ice layering, and then collect samples for cell counts of various basal ice facies at regular intervals. Total cell counts are easy to do and cheap, so there are no obvious reasons to restrict the

number of samples to just six samples. The reasoning behind the sampling strategy and selection of sampling sites needs to be explained, so that it is clear to the readers why the applied sampling strategy is better than sampling along profiles and why six samples are sufficient to estimate the abundance and variability of cells in basal ice.

2,29: Insert "assumed to be" before "similar".

3,2-3: Repetition. Delete this sentence.

3,15 and below: I think that it will be more logic to present the calculation method (3,27-4,2) before writing about the conditions at Svinafellsjökull. Therefore, I will suggest that the authors consider switching the two paragraphs in section 2.4.

3,16: Repetition. Delete this sentence.

3,16-25: This is very central for the calculations and the associated uncertainty estimates, but unfortunately the explanation presented in this paper is not very clear to me. I think that the calculations and assumptions by Cook et al. (2010, 2011b) should be presented in much more detail and with better descriptions of the estimates of each variable. It should not be necessary for readers to consult the two papers by Cook et al. to understand, for example, what is meant by stratified ice formed by glaciohydraulic supercooling. What is the length of the basal ice exposure around the glacier margin? How was the length measured? Was the length corrected because of glacier retreat between the study of Cook et al. (2010) and the present study? What is the ablation rate and ice velocity at the glacier front?

3,25-26: What is the spatial distribution (in %) between the different ice facies? How what it estimated and what are the uncertainties?

3,27: Delete "m-1".

4,5-6: It is difficult to assess these results without information of differences in grain size distribution and the content of content of gravel and larger particles.

4,7: Delete extra spacing.

4,21: Separate the Discussion section from the Conclusions section.

4,23-26: This comparison with other cell counts from basal ice facies from glaciers must be expanded and discussed in detail in relation to environmental differences and similarities (e.g., lithology, basal thermal regime, basal ice facies, debris concentration and grain size distribution).

4,28-29: This comparison with cell counts from supraglacial, glaciofluvial and terrestrial proglacial environments also needs to be expanded and discussed in context to deliver of cells from basal ice to adjacent environments.

5,5: Delete "bacterial".

5,7-10: Repetition. Delete these sentences.

5,12: What is meant by "who also found that cell counts increased with sediment content"? This relationship is not determined in the present paper.

5,13-14: "It is clear that different ice facies deliver different amounts of cells to the glacier margin". Why is this clear? I thought that the main conclusion from this study (based on six samples) was that stratified and dispersed ice facies contained similar amount of cells per gram of debris, making the distribution of different basal ice facies an insignificant variable. The main control on cell flux would then be the debris concentration in basal ice. Have I misunderstood something? If not, I think that this sentence should be rephrased to emphasize that debris concentration is the important parameter and that there is no need to consider various basal ice facies.

5,18-22: Again, this discussion of the role of different subglacial factors needs to be expanded and include a proper literature analysis of the microbiology in basal ice rather than being restrict to a single reference.

5,22-23: "Hence, we recommend that similar studies be performed at other sites with

different glaciological characteristics to gain a better application of cell transfer to the margins of glacier". Where it is possible, cell delivery should be calculated from existing debris fluxes and cell counts from other glaciers, and the results should be compared with the results from Svinafellsjökull. Based on this comparison, the authors may recommend more studies on cell delivery from basal ice.

7,6: Is there a name missing here?

Figure 1a: Difficult to read the text at the bottom of Figure 1a. Is this text necessary?

Referred literature not mentioned in the paper:

Hunter et al.: Flux of debris transported by ice at three Alaskan tidewater glaciers, J. Glaciol., 42(140), 123-135, 1996.

Irvine-Fynn and Edwards: A frozen asset: The potential of flow cytometry in constraining the glacial biome, Cytometry A, 85(1), 3-7, 2014.

Knight et al.: Discharge of debris from ice at the margin of the Greenland ice sheet, J. Glaciol., 48(161), 192-198, 2002.

Yde et al.: Basal ice microbiology at the margin of the Greenland ice sheet, Ann. Glaciol., 51(56), 71-79, 2010.

---

## Author Comment (AC1) · 27 Jan 2017

We thank Reviewer 1 for their constructive and critical review of our manuscript. There are some valuable observations in the review that will lead to improvements in our revised manuscript. We take this opportunity to respond to comments and highlight what changes we intend to make in the revised manuscript. Our replies to each comment begin with "»>".

1) Claims of cell discharge relate to total cells derived from DAPI counts and viable cells from CFU counts obtained from the inoculation of a specific growth medium under one set of incubation conditions for five weeks. While total counts from microscopy are acceptable, I firmly disagree with the notion that the authors have determined viable counts, and whether the procedures employed are adequate to answer their exnone

perimental question on the following grounds: A: Viable does not mean culturable. Consider the very paradox of the acronym "Viable But Not Culturable" which has been explored for >30 years by many investigators. As the issue of VBNC sets out, one of the challenges of contemporary microbial ecology is understanding the gap between who appears on your agar plate (culturable), who is present (total, including dead cells) and who might live in situ (viable) and those who are able to live in situ but not on your agar plate (VBNC). The paper needs to take into account that viability is non synonymous with culturability. Here culturability under one set of conditions is presented. This means very little for quantitative estimation of viability. If one is minded to determine the abundance of viable cells within an environmental habitat, very different tools are required, typically in the vein of Live/Dead stains and microscopy. These are not without their problems of course. B: One set of conditions are tested: 10% TSA, 4 deg C for five weeks. No data is presented setting out whether this set of conditions is representative or optimal. What assurance does the reader have that this protocol provides consistent counts? C: So if viability itself is not quantified what about culturability itself - is it meaningful? What does growth in vitro really tell us about those cells' ability to colonize proglacial environs? I believe it was the eminent microbiologist John Postgate who stated that "every colony is an artefact". It is difficult to convincingly aruge that culture of cells in vitro under one set of fixed conditions necessarily provides quantitative insights to the in situ actuality. D: No information is provided on the community composition of the inhabitants of the basal ice or the proglacial habitats they may be discharged into. Clearly, not all microbes have the same potential to colonize an environment. The ecological impact of inoculating a trillion cells which cannot persist and grow in the forefield will be very different to just one cell immured in basal ice which can also thrive in the forefield. As such, the implications of mass transfer of cells are poorly developed, and assume equivalency of outcome across what are very different scenarios: are these cells likely to pioneer the forefield community's development, or are they simply a source of nutrients as necromass? Or will dormant cells provide a long term repositry of genetic potential for later stages of soil development? Very different ecological scenarios arising from the physiological state and colonization potential of the source microbiota which are beyond the scope of the analyses performed. As such I feel the development of the rationale of this underlying motivation of the paper is limited in its grasp, and the paper would really benefit from careful consideration of the processes underlying the assembly of microbial communities.

»>The reviewer is correct - we used the term "viable", which is rather old-fashioned and inaccurate terminology, and therefore misleading. We will replace the term with "cultivable", in the revised manuscript. The reviewer also states that the cultivation technique is inadequate to answer our experimental question. However, the rationale for carrying out cultivable, in addition to total, counts was given in the methods as "In order to analyse if the community inhabiting the basal ice supported alive and viable. . .". Environmental microbial cultivation, using classical dilution plating on diluted TSA, obviously does not provide an absolute quantification of the viable microbial load in the glacier and we do not claim this to be the case in our study (excepting the error of terminology noted above). In any case, we distinguish the difference between cultivable and viable cells in the text and Figure 2, in presenting the cultivable counts as "conservative estimates of the delivery of viable cells to the ice margin". We stand by this – what we are presenting is a lower limit of viable cells and there is plenty of scope for future work to refine our estimates to further highlight the significance of basal ice derived microbial cell discharge. There is no guarantee that cultivation with 10 % TSA is "optimal" but there is also no requirement for it to be optimal and no reason to expect it not to be "representative" – at least for the purpose of comparisons between our samples. A look at several studies where microorganisms have been recovered from cold/icy environments reveals that there have been a range of different isolation media used (e.g. R2A e.g. Montross et al., 2014 , TSA e.g. Miteva 2004, LB e.g. Shivaji et al., 2011 and even other media, Foght et al., 2004), and with different media strengths (2%, 10%, 50%, 33%). Our purpose was simply to provide support for the microscopic counts, plus indicating the presence of viable cells. The reviewer seems to think we had a greater purpose than this, which is probably because we used in-
accurate terminology (now removed). As the reviewer points out, to properly quantify the viable fraction of cells in the glacier is no trivial task, and as such it is way outside the scope of this work. Assessing microbial community composition and community assembly was also outside the scope of this work, in which we chose to focus on microbial cell transfer rates as a discrete subject that is clearly identifiable in the title of the paper. The underlying motivation of the paper is, we believe, clearly stated and not "limited in grasp" as suggested by the reviewer. The focus is a first assessment of transfer of microbial cells in the basal glacial environment and not their taxonomy. We will perform detailed geo-microbial community analyses of basal ice using culture-dependent and culture-independent methodology that will be reported separately. The results presented in the current study will inform interpretation of functional basal ice geo-microbiology data.

2) At the heart of this paper are total counts and CFU counts. While their use coupled with expert interpretation of the basal ice facies is important, this seems a little preliminary and the conclusions drawn risk superficiality as a result. The paper would be greatly strengthened as an offering to the literature if it described the taxonomic composition of the cultured and total community. As noted above, simply dumping cells into an environment has radically different outcomes dependent on the identity of the cells.

»>Yes, the heart of this work is total counts and CFU counts – coupled with glaciological data about the nature of ice facies, included sediment, and sediment discharge. As mentioned above, this provides a discrete result and this was our aim. The findings are relevant to both the glaciology community and the microbiology community. As stressed above, to describe exhaustively microbial taxonomic composition in basal ice is outside the scope of the current study, and warrants reporting separately in a follow up microbial diversity focused paper. We also wanted to make the point in our paper that there are key differences between ice types, which might affect the microbiology – this is usually overlooked in glacial microbiology studies, and in our opinion worthy of discussion. So we feel our results and message are somewhat broader than suggested

above by Reviewer 1.

3) How representative is this site of other locations? I appreciate that its history of circumspect glaciological investigation lends itself for this study, but considering its history of advance over soils, what lessons can be learned from this site that would be applicable to sites with very different histories?

»>As the reviewer states, a key advantage of working at this site is that the sediment transfer system is well known to the authors – in particular, Cook has worked here for over a decade, and published a number of studies on sediment transfer at Svínafell-sjökull. But the reviewer raises an important point about the extent to which this site is representative of other glacial systems more generally. We suggest that this site is representative of many of the glaciers in this region, and could be considered representative of other temperate glaciers elsewhere. Of course, we acknowledge that there can be significant variability between glaciers, even of the same thermal regime, size, prevailing climate, etc. Svínafellsjökull is a temperate valley glacier that has experienced periods of advance and recession contemporaneously with other glaciers in southern Iceland (Hannesdottir et al., 2015). During recession, there has been opportunity for soil and vegetation development in the area currently occupied by glacier ice. This is true of many of the glaciers in this region – for example, Ives (2007) highlights documentary reports from neighbouring Skaftafellsjökull where the glacier had once been mined for birch wood overridden by the glacier. Other glaciers globally have experienced similar patterns of recession and advance during the Holocene. Given that glaciers commonly experience phases of recession and advance, we suggest that Svínafellsjökull is a representative site at least regionally, and potentially globally for other valley glacier systems. In our revised manuscript, we will highlight how our results are applicable more generally.

4) L8: "We present the first assessment of microbial cell discharge from sediment laden glacier basal ice." L28: "We report the first quantification of microbial discharge to a glacier margin, and demonstrate that there is viable microbial inoculum released to the

proglacial environment"

Respectfully, I disagree with the assertion of priority made for this claim, and the emphasis provided by placing it at the start of the abstract. Starting from the seminal paper of Sharp et al (1999) microbial prevalence in basal ice has been widely documented as has its potential for inoculating forefields as well as demonstratable culturable bacteria using a range of methodologies, as well as culture-independent strategies (e.g. Kaštovská et al 2007; Yde et al 2010 Ann Glaciol; Montross et al 2015 Geomic J; Rime et al 2016 ISMEJ). I would highly recommend a more circumspect statement regarding the motivation of this study which clearly and fairly asserts the scientific novelty of the work. Perhaps the emphasis of integration with basal ice extent is required?

»>The reviewer is correct in stating that we are not the first researchers to investigate microbial prevalence in basal ice, nor are we the first to document the potential for basal ice microbes to promote proglacial soil and vegetation development. We will re-word the abstract and the Introduction in the revised manuscript to reflect the fact that there has been literature to suggesting that microbial inocula are important for proglacial ecosystems. We now recognise that the way in which we had written this appeared to be disingenuous. The reviewer is also correct in stating that it is the combination of our microbiological data with the glaciological/geographical data about basal ice thickness, sediment content and extent that adds novelty to our study. We see no basis for the reviewer's first criticism here about us not being the first researchers to examine microbial prevalence in basal ice – we never make such a claim. The point of our paper is that we are the first researchers to quantify the discharge of microbes from basal ice – not to be the first to quantify microbial prevalence in basal ice. We feel that we have already been clear about this throughout, including in the statements cited by the reviewer above.

5. L16: The authors emphasize the heterogeneity inherent to these ice facies. The methods section does not set out how the samples collected were distributed across the ice facies to describe this hetereogeneity and minimise potential biases. In short,

what was the specific survey design, and the extent of replication. Fig1a goes some way to explain the number of sites sampled, but more clarity is needed here, especially on potential intra-site variation.

»>Yes, the heterogeneity in basal ice is a key point that we want to get across to those interested in sampling basal ice for microbes. It was for this reason that we developed a targeted basal-ice sampling methodology described in detail in Toubes-Rodrigo et al. (2016), which is cited in the text. Ultimately, we think the reviewer has slightly misunderstood the aims of our study, but this has prompted us to make some minor clarifications to the text. Briefly, previous studies of basal ice microbiology, with the exception of Yde et al. (2010) and Montross et al. (2014), have not accounted for the fact that basal ice typically comprises different ice types/facies of different origins and characteristics. Our point really is that it is too crude to state that one sampled "basal ice" that is likely to be comprised of different ice facies, that could potentially explain differential glacier-specific microbial content. Our study acknowledges that the two basal ice types, stratified facies and dispersed facies, exhibited different physical characteristics. Whilst there might be slight differences between, for example, a piece of dispersed facies sampled from the north of the glacier versus a piece that is extracted from the south, the samples are from descriptively the same ice type with the same origins – as demonstrated by Cook et al. (2007, 2010, 2011a). We will provide photos of the sampled sites to try to clarify this in the revised manuscript, and will clarify the lack of intra-site variation in the text.

6. Uncertainties in sediment transfer rates. These seem pretty broad, and incur a two-fold variation in the potential discharge of cells. Can the authors justify the insights afforded by this calculation considering this considerable uncertainty?

»>Our cell discharge values are based on glacier velocities ranging from 4 to 8 cm/day (or 14.6 to 29.2 m/year), as reported in Cook et al. (2010). These data were acquired over a 4-week survey period in Summer 2007 using a total station. We have undertaken subsequent satellite remote sensing work (feature tracking), which confirms velocities

in the terminus to be <40m/year, averaged over a year from 2002 to 2003. It is perhaps unfortunate not to have convergence on a closer range in velocity, and hence, cell discharge values. We could instead have reported the mean velocity rather than the range, but decided it best to be transparent about the range of values observed. In reality, it is likely that velocity changes rather a lot as subglacial hydrological networks evolve seasonally, for example. We think the range in values is a better way to reflect this natural variability, which is important because this is the first time anyone has attempted to quantify cell discharge from basal ice. We will clarify this in the revised manuscript. We also argue that we are making an important separate point about cell release from different basal ice facies at this glacier, which is a function of sediment content and basal ice extent and thickness. As we have discussed, known basal ice heterogeneity is an important justification for the rationale of our work and it is the first time anyone has examined variations in cell discharge between different basal ice facies.

7. How does basal ice microbial discharge scale up relative to fluxes from meltwater or till? Context could be provided here.

»>Frankly, we don't know! This is something that needs to be addressed by the glacial microbiological community. This could be relatively straightforward to assess for subglacial meltwater – ideally, one would want a glacier where the majority of the water is discharged at the glacier front through a single portal. Irvine-Fynn and Edwards (2014) estimated that $3.2 \times 10^{21}$ cells a-1 were discharged from glacial meltwater systems worldwide. Studies on cell discharge from till would be more challenging. There is still much disagreement on the pervasiveness and depth of till deformation, and how to model its flow (e.g. Clarke, 2005), and instrumenting the till in a representative way to gain insights into rates of movement would be difficult. Certainly, we can consider adding some context to our results, but it is difficult to address the reviewer's comment more fully due to the lack of available data from other studies.

8. Discussion needs to draw out the insights into the ecological processes affected by

cell discharge from basal ice. What does it all mean for the downstream habitat?

»>This paper does not directly lead to ecological insight. Rather, it provides data that can support such insight in future work - for example, our own microbial community analysis mentioned earlier. We believe that the addition of ecological interpretation is out of the scope of this publication. The aim of this paper is to present a calculation of microbial delivery from basal ice to the glacier margin.

MINOR COMMENTS.

L27: More detail is needed here on sampling protocol and precautions to allow readers without access to the cited source to evaluate the protcol applied.

»>Thanks. Yes, we will add more detail in the revised manuscript.

L29: Ballpark figure provided by Shivaji et al (2011). Microbial abundance changes considerably as soil develops over a chronosequence. Perhaps your basal ice abundances matter more when meeting the depauperate bare till of the immediate glacier margin.

»>Yes, that's a good point.

L30: Formamide? or formaldehyde? The reader needs to be reassured the microbial population is adequately fixed for enumeration. & L32:

»>Formaldehyde in both cases.

L36: Using DAPI on small and dormant cell populations. What assurances does the reader have about the sensitivity of DAPI in this context given its lower quantum yield of fluorescence relative to SYBR stains?

»> The use of DAPI is a well-established technique, and it has been used previously to investigate basal ice microbiology (Yde et al., 2010). For valid comparisons the same concentration has been applied in this study.

REFERENCES

Clarke, GK.C. (2005) Subglacial processes. Annual Reviews of Earth and Planetary Science, 33, 247-76.

Cook, S. J., Knight, P. G., Waller, R. I., Robinson, Z. P. and Adam, W. G.: The geography of basal ice and its relationship to glaciohydraulic supercooling: Svínafellsjökull, southeast Iceland, Quat. Sci. Rev., 26(19–21), 2309–2315, 2007.

Cook, S. J., Robinson, Z. P., Fairchild, I. J., Knight, P. G., Waller, R. I. and Boomer, I.: Role of glaciohydraulic supercooling in the formation of stratified facies basal ice: Svínafellsjökull and Skaftafellsjökull, southeast Iceland, Boreas, 39(1), 24–38, 2010.

Cook, S. J., Swift, D. A., Graham, D. J. and Midgley, N. G.: Origin and significance of "dispersed facies" basal ice: Svínafellsjökull, Iceland, J. Glaciol., 57(204), 710–720, 2011a.

Cook, S. J., Graham, D. J., Swift, D. A., Midgley, N. G. and Adam, W. G.: Sedimentary signatures of basal ice formation and their preservation in ice-marginal sediments, Geomorphology, 125(1), 122–131, 2011b

Foght, J., Aislabie, J., Turner, S., Brown, C. E., Ryburn, J., Saul, D. J. and Lawson, W.: Culturable bacteria in subglacial sediments and ice from two Southern Hemisphere glaciers., Microb. Ecol., 47(4), 329–340, 2004.

Hannesdóttir, H., Björnsson H., Pálsson F., AÃřalgeirsdóttir G., GuÃřmundsson, Sv.: Changes in the southeast Vatnajökull ice cap, Iceland, between ∼ 1890 and 2010, Cryosph., 9(2), 565–585, 2015.

Irvine‐Fynn, T. D. L. and Edwards, A.: A frozen asset: the potential of flow cytometry in constraining the glacial biome, Cytom. part A, 85(1), 3–7, 2014.

Ives, J. D.: Skaftafell in Iceland: A Thousand Years of Change, Ormstunga, 2007.

Miteva, V. I., Sheridan, P. P. and Brenchley, J. E.: Phylogenetic and physiological diversity of microorganisms isolated from a deep Greenland glacier ice core, Appl. Environ. Microbiol., 70(1), 202–213, 2004.

Montross, S., Skidmore, M., Christner, B., Samyn, D., Tison, J. L., Lorrain, R., Doyle, S. and Fitzsimons, S.: Debris-Rich Basal Ice as a Microbial Habitat, Taylor Glacier, Antarctica, Geomicrobiol. J., 31(1), 76–81, 2014.

Shivaji, S., Pratibha, M. S., Sailaja, B., Hara Kishore, K., Singh, A. K., Begum, Z., Anarasi, U., Prabagaran, S. R., Reddy, G. S. N. and Srinivas, T. N. R.: Bacterial diversity of soil in the vicinity of Pindari glacier, Himalayan mountain ranges, India, using culturable bacteria and soil 16S rRNA gene clones, Extremophiles, 15(1), 1–22, 2011.

Toubes-Rodrigo, M., Cook, S. J., Elliott, D. and Sen, R.: 3.4. 1. Sampling and describing glacier ice, in Geomorphological techniques, edited by Cook, S.J., Clarke, L.E. Nield, British Society for Geomorphology, London., 2016.

Yde, J. C., Finster, K. W., Raiswell, R., Steffensen, J. P., Heinemeier, J., Olsen, J., Gunnlaugsson, H. P. and Nielsen, O. B.: Basal ice microbiology at the margin of the Greenland ice sheet, Ann. Glaciol., 51(56), 71–79, 2010.

---

## Author Comment (AC2) · 27 Jan 2017

Reviewer 2 provided a helpful and critical review of our manuscript, for which we are grateful. Many of the comments appear to be aimed at encouraging us to expand our manuscript by adding details in a number of areas (e.g. about the site, sampling strategies, comparisons with other studies). We discuss below how Reviewer 2's comments have led to improvements in our revised manuscript. Our responses start with "»>".

1. The first thing that attracts my attention is that the paper is unnecessarily short for a research paper in Biogeosciences. It gives the impression that the paper was intended for another journal but somehow ended up as a submission to Biogeosciences. As the paper addresses a very specific topic within subglacial microbiology, namely total cell counts and cell flux in basal ice, I strongly recommend that the paper is expanded

to provide the readers with an up-to-date overview of the current knowledge of cell abundance in basal ice and place the new cell counts and fluxes in this context. This will undoubtedly increase the impact of the paper.

»>This is a fair point. We had deliberately tried to keep our manuscript short with the original intention that it would get our point across succinctly. When writing inter-disciplinary manuscripts (microbiology, glaciology, sediment systems in our case), it can be challenging to present the work in a succinct manner whilst also maintaining transparency and comprehensibility. However, since both reviewers' comments have requested more detail, we are happy to expand the paper to enhance understanding and potential impact of the work. By addressing the comments of both reviewers, this manuscript will expand significantly. We hope, therefore, that we have addressed this particular comment from Reviewer 2.

2. The rationale for the study is that "basal ice melt-out could deliver viable micro-biota to the ice margin that serve as inoculum, potentially accelerating pedogenesis as glaciers recede" (1,24-25). This may be true for some glaciers, but when I look at the location map (Figure 1a) it seems that the entire front of Svinafellsjökull is in contact with either a glacial river or ice-marginal lakes. Hence, the cells that melt out at the six sampling sites will most likely be washed into the glacial river and transported to a downstream sandur or into the sea. It is unlikely that they will accelerate pedogenesis as Svinafellsjökull recedes. As this is a case study, the scientific rationale should reflect what is relevant for the environment at Svinafellsjökull. I suggest that the authors put more emphasis into presenting the ice-marginal environment and a rationale that addresses the conditions at Svinafellsjökull. This will also make the paper more interesting for potential future studies on the microbial community structures in the supraglacial environment, the basal ice, the proglacial river/lakes, and the proglacial foreland at Svinafellsjökull.

»>The reviewer is correct in asserting that there are, at present, a number of proglacial water bodies at Svínafellsjökull. However, this has not always been the case – for example, aerial photos from 1994 illustrate that most of the glacier was in contact with moraine, so the glacier foreland at that time may have received significant contributions of basal ice microbes. The contact zone between the glacier and proglacial area is dynamic. What we hope is that our paper begins the process of quantifying cell discharge from basal ice, and makes the point that this could be important for soil development. Hopefully, future studies will allow similar quantification from other sites with differing dynamic conditions. Basal ice melting out from the current glacier margin could contribute microbial material to moraines and ice-marginal sediment accumulations where it could accelerate pedogenesis, but we agree that in many areas microbes will be transported away by rivers or stored in lakes. Again, the implications of microbial loss/storage and contributions to pedogenesis have barely been studied, hence the novelty of our study. We hope that much more will be done in the future. We will add extra information to the manuscript about the dynamics of the glacier forefront. In addition, we will include a new figure of the sampling sites.

3. The Introduction section is basically written as "there is a lot of knowledge about this, but little knowledge about that". This form is not very interesting to read and it seems a bit dubious at times. For instance, the authors write that "few studies have quantified sediment discharge from basal ice . . . (Wainwright et al., 2015)" (1,29-20), whereas Wainwright et al. (2015), in fact, write that "several studies [e.g., Hunter et al, 1996; Knight et al., 2002] have measured actual debris flux through the basal layer" (see page 1182 in Wainwright et al., 2015). I recommend that the authors change the form and include many more relevant references and use them in an active way (e.g., "Cook et al. (2010) found that . . ."). A full literature overview of cell counts conducted on different basal ice facies would be relevant either in the Introduction section or the Discussion section (maybe as a table).

»>We argue that introduction sections often comprise elements of literature review – what is the state of the science? What do we know? What don't we know? What will the present study do to address what we don't know? That is a common format that

we are content with.

With respect, we disagree with the implication that several studies of basal ice sediment discharge/flux have been undertaken. The use of the word "several" by Wainwright et al. (2015) is, in our view, unfortunate. Take the most recent of the 2 example references used in that quote, i.e. the reference to Knight et al. (2002) – in that paper it is stated that:

"relatively little work has examined how the flux of debris through the basal ice layer contributes to glacial sediment budgets".

And,

"...their review revealed...the limited information about debris flux through the basal ice layer"

Indeed, it is fair to say that the dearth of information on sediment fluxes from basal ice was the very rationale for the paper by Knight et al. (2002).

A few other papers have been published since 2002 on sediment fluxes through basal ice – those of Cook et al. (2010, 2011a) and Larson et al. (2006). Indeed, one of the current authors (Cook) is writing a book and a separate manuscript on this very subject highlighting how few studies have been undertaken. All of this does rather depend on one's definition of the word "several", as used in Wainwright et al.'s quote above. In our view, this should say "few". Compare the number of studies of sediment flux through basal ice to the number of studies on sediment flux through rivers, or glacial rivers even, and one will find that such studies are extremely rare, unfortunately. This is one of the reasons why Svínafellsjökull is an ideal study site – Cook et al (2010, 2011a) quantified sediment discharge here. So we contest the reviewer's point that our Introduction section is "a bit dubious at times".

We do, however, concede that more references could have been used. As we explain above, we had tried deliberately to keep this manuscript short, but on the recommendation of the reviewer (both reviewers, in fact), we will several references to the introduction. Also a new table will be added in which cell counts from different studies have been included.

4. It is stressed out throughout the paper that this is the 'first' quantification of cell flux from basal ice. If it is so important to provide the first quantification of cell flux from basal ice, the authors could just have combined the debris flux in basal ice provided by Knight et al. (2002) with the total cell count in basal ice provided by Yde et al. (2010) to produce an estimate for the basal ice delivery of cells at Russell Glacier in Greenland. This will have saved them all the fieldwork. Although it may be true that this first to make this estimation, I will suggest that the mentioning about being the 'first' paper to quantify basal ice cell flux is toned down and replaced by quantitative and qualitative comparisons between the results from Svinafellsjökull and estimates of debris fluxes and cell counts from other glaciers.

»> The proposal to combine data from Knight et al. (2002) and Yde et al. (2010) does not seem very feasible. Knight et al. (2002) quantify sediment discharge across a stretch of the ice margin, which comprises both dispersed and stratified facies ice – the stratified ice is further divided into 3 sub-facies (solid, discontinuous, suspended). The study by Yde et al. (2010) describes a range of ice types, but the debris-bearing ice types are banded facies and solid facies (importantly, dispersed facies was not observed or sampled despite it accounting for around 1/3 of the sediment discharge reported in Knight et al., 2002). Cell counts are reported only for the solid facies by Yde et al. (2010), and not the other 2 sub-facies of the stratified facies. Given that Knight et al. (2002) found that solid facies accounts for only $\sim$56% of the sediment discharge from the basal ice layer, it would not be possible to present a full story of the microbial cell discharge from the basal ice at that site by combining the datasets from these studies.

Most studies try to outline what it is about the work that constitutes originality. One of the original elements of our work is that, unlike the problems outlined above, we can for

the first time present an integrated glaciological and microbiological dataset that allows us to estimate cell discharge. This is the first time that this has been attempted, and we would like to be able to continue to highlight this as an original aspect of our work.

MINOR COMMENTS

»>Thanks for spotting these!

1,9 and 1,10: These numbers of cell flux and cell abundance in the Abstract should be similar to numbers found in the Results section

»>CHANGED

1,17; 1,19; 1,20; 1,30; 1,31-2,1 and other places: Include more references to support these statements. »>ADDED 1,20 and other places: Insert comma after "et al."

»>ADDED

1,25: A second paragraph should provide a literature overview on microbial abundance in glacier ice, including basal ice (see e.g. Irvine-Fynn and Edwards, 2014). There is no need to go into details about basal ice microbial diversity, expect for where the microbial diversity is relevant for culturing of cells. A third paragraph could be on debris fluxes from basal ice.

»> We will these new paragraphs. However, we think that the text flows better if we reverse the order and add a first paragraph on sediment fluxes and a second one on the microbiology of basal ice.

2,1-3: "Our aims were to . . . and confirm that viable microbial inoculum are transferred between glaciers and proglacial ecosystems". It is well known that viable cells are transported by subglacial river to the proglacial environment, so it must be specified that the aim of the paper is focusing on basal ice transport of microbes to a fluvial proglacial ecosystem.

»>We do now mention in the text the geomorphological setting. Nonetheless, there

are still sizeable areas where the glacier is contact with the moraine/till, and hence microbes can contribute to soil formation. As we discuss above, this has not always been the case – in the past, a much greater proportion of the foreland was dominated by discharge to till/moraine, so the results of the present study still have general applicability under these circumstances.

2,7-12: The site description must provide more relevant information, especially regarding the proglacial river and ice-marginal lakes. What is the meteorological regime? What is the distance of glacial retreat since the Little Ice Age? What is the contemporary average frontal retreat rate per year? What is the river discharge and suspended sediment load? Is anything known about the supraglacial or proglacial microbial communities?

»> Certainly, we can add some of these sorts of details to provide additional context about the site. However, some of this information seems to us to be somewhat superfluous. Again, the geomorphological conditions can change dramatically over time, but the point remains: microbial discharge has contributed significantly to proglacial soils in the past, and continues to do so where the ice is in contact with till/moraine. Certainly, we can clarify the nature of these changing conditions in our revised manuscript.

2,9: The period mark is red.

»>SOLVED

2,18 and 2,23: With regards to debris content, it is more correct to use "by mass" instead of "by volume" and to present supplementary information on grain size distributions. This will also be consisted with the use of mass in the calculations (3,28-29). If the stratified or dispersed facies contain gravel, stones or boulders, it should also be noted.

»>Different measures have been used to describe sediment content in basal ice in different studies. Both these data, and data pertaining to particle size distributions
have been reported elsewhere (Cook et al., 2010, 2011a,b). We will, however, give careful consideration about what information to include here.

2,19: Insert "a" before "layer".

»>ADDED

2,25: How far up-glacier is the icefall?

»>ADDED

2,26: I don't understand the term "strain-related metamorphophism". Isn't all change of ice crystals in solid-state (i.e. metamorphosis) strain-related?

»>Pervasive grain-boundary melting and refreezing (e.g. Tison and Hubbard 2000). This is a commonly used term (e.g. Hubbard and Sharp, 1995).

2,27-28: In my opinion, the most obvious sampling strategy would have been to select and survey one to three basal ice profiles perpendicular to the basal ice layering, and then collect samples for cell counts of various basal ice facies at regular intervals. Total cell counts are easy to do and cheap, so there are no obvious reasons to restrict the number of samples to just six samples. The reasoning behind the sampling strategy and selection of sampling sites needs to be explained, so that it is clear to the readers why the applied sampling strategy is better than sampling along profiles and why six samples are sufficient to estimate the abundance and variability of cells in basal ice.

»> In an ideal situation it would have been great to have undertaken sampling in the manner described by Reviewer 2. At Svínafellsjökull, basal ice is not always exposed or well-exposed, and in many instances is not safe to access. During the 2015 campaign, three places showed conveniently accessible stratified facies for sampling, although this facies was observed in several locations. We were also driven to sample dispersed facies from the same locations (example, S3-D3) in order to be able to compare directly between ice types. However, this placed a further constraint on sampling.

3,15 and below: I think that it will be more logic to present the calculation method (3,27-4,2) before writing about the conditions at Svinafellsjökull. Therefore, I will suggest that the authors consider switching the two paragraphs in section 2.4.

»>Respectfully, we disagree. We think that it would be better to start with the conditions at Svínafellsjökull since previous studies have calculated debris discharge introducing the way we have performed our calculation. Also, to clarify this point the equation formulae for sediment and cell discharge will be included.

3,16: Repetition. Delete this sentence.

»>DONE

3,16-25: This is very central for the calculations and the associated uncertainty estimates, but unfortunately the explanation presented in this paper is not very clear to me. I think that the calculations and assumptions by Cook et al. (2010, 2011b) should be presented in much more detail and with better descriptions of the estimates of each variable. It should not be necessary for readers to consult the two papers by Cook et al. to understand, for example, what is meant by stratified ice formed by glaciohydraulic supercooling. What is the length of the basal ice exposure around the glacier margin? How was the length measured? Was the length corrected because of glacier retreat between the study of Cook et al. (2010) and the present study? What is the ablation rate and ice velocity at the glacier front?

»>We will add further details about this – thanks.

3,25-26: What is the spatial distribution (in %) between the different ice facies? How what it estimated and what are the uncertainties?

»>This has been reported previously in Cook et al. (2010). We can add details about this in the revised manuscript.

4,5-6: It is difficult to assess these results without information of differences in grain size distribution and the content of content of gravel and larger particles.

[Figure]

»>We are not entirely sure what the rationale is here. Grain size distributions have been presented elsewhere (Cook et al., 2010, 2011a,b).

4,23-26: This comparison with other cell counts from basal ice facies from glaciers must be expanded and discussed in detail in relation to environmental differences and similarities (e.g., lithology, basal thermal regime, basal ice facies, debris concentration and grain size distribution).

»> This is a fair point, but in practice is rather difficult to achieve. A key point that we make here is that previous studies on glacial microbiology fail to adequately describe basal ice types, making it difficult to draw comparisons. We will consider this point, however, and add any further pertinent details and comparisons that we can.

4,28-29: This comparison with cell counts from supraglacial, glaciofluvial and terrestrial proglacial environments also needs to be expanded and discussed in context to deliver of cells from basal ice to adjacent environments.

»» The aim of this research is not addressing the relationship between supra and sub-glacial environment or between basal ice and glaciofluvial or glaciolacustrine environments. We could expand the information about the presence of microorganisms in the glacier foreland, because it would be relevant for this work.

5,5: Delete "bacterial"

»>This number corresponds exclusively to bacterial counts, since it pertains to the counts of CFU in 1:10 TSA with cycloheximide, which prevents the growth of fungi.

What is meant by "who also found that cell counts increased with sediment content"? This relationship is not determined in the present paper.

»>Montross et al. (2014), found that the number of cells in the ice increased with the sediment concentration. Our results indicate that microorganisms are more abundant in the sediment entrapped in the ice (stratified is debris-rich, dispersed is debris-poor).

5,13-14: "It is clear that different ice facies deliver different amounts of cells to the glacier margin". Why is this clear? I thought that the main conclusion from this study (based on six samples) was that stratified and dispersed ice facies contained similar amount of cells per gram of debris, making the distribution of different basal ice facies an insignificant variable. The main control on cell flux would then be the debris concentration in basal ice. Have I misunderstood something? If not, I think that this sentence should be rephrased to emphasize that debris concentration is the important parameter and that there is no need to consider various basal ice facies.

»>Basal ice facies are commonly described and differentiated by sediment content – different basal ice facies deliver different amount of cells to the glacier margin as a function of the different sediment content. As Reviewer #2 has highlighted that our explanation was not clear enough and for the final manuscript we will re-phrase in the text to make it more understandable.

5,18-22: Again, this discussion of the role of different subglacial factors needs to be expanded and include a proper literature analysis of the microbiology in basal ice rather than being restrict to a single reference.

»>We cite one reference here because Lawson et al. (2015) is the only study that is similar to ours (i.e. which quantifies cell concentration). We could speculate about how other factors might affect microbial ecology here, but much of it probably falls beyond the focused remit of our study. We would be happy to add a new table comparing cell content in basal ice in different glaciers if wanted.

5,22-23: "Hence, we recommend that similar studies be performed at other sites with different glaciological characteristics to gain a better application of cell transfer to the margins of glacier". Where it is possible, cell delivery should be calculated from existing debris fluxes and cell counts from other glaciers, and the results should be compared with the results from Svinafellsjökull. Based on this comparison, the authors may recommend more studies on cell delivery from basal ice.

»> We are not sure if we have understood this comment correctly, but we think this is similar to an earlier point about integrating data from other studies. We outlined earlier, in perhaps the best studied site (Russell Glacier), that there are large gaps in the dataset that would need to be filled in order to derive these cell discharges. It is a good idea but we are not aware of suitable data to make these calculations with any conficene.

7,6: Is there a name missing here?

»> No, but we do need to remove the "J." – thanks for spotting this.

Figure 1a: Difficult to read the text at the bottom of Figure 1a. Is this text necessary?

»>That text is accrediting the source of the image

REFERENCES Cook, S. J., Robinson, Z. P., Fairchild, I. J., Knight, P. G., Waller, R. I. and Boomer, I.: Role of glaciohydraulic supercooling in the formation of stratified facies basal ice: Svínafellsjökull and Skaftafellsjökull, southeast Iceland, Boreas, 39(1), 24–38, 2010.

Cook, S. J., Swift, D. A., Graham, D. J. and Midgley, N. G.: Origin and significance of "dispersed facies" basal ice: Svínafellsjökull, Iceland, J. Glaciol., 57(204), 710–720, 2011a.

Cook, S. J., Graham, D. J., Swift, D. A., Midgley, N. G. and Adam, W. G.: Sedimentary signatures of basal ice formation and their preservation in ice-marginal sediments, Geomorphology, 125(1), 122–131, 2011b.

Hubbard, B. and Sharp, M.: Basal ice facies and their formation in the western Alps, Arct. Alp. Res., 301–310, 1995.

Knight, P. G., Waller, R. I., Patterson, C. J., Jones, A. P. and Robinson, Z. P.: Discharge of debris from ice at the margin of the Greenland ice sheet, J. Glaciol., 48(161), 192–198, 2002.

Larson, G. J., Lawson, D. E., Evenson, E. B., Alley, R. B., Knudsen, Ó., Lachniet, M. S. and Goetz, S. L.: Glaciohydraulic supercooling in former ice sheets?, Geomorphology, 75, 20–32, 2006.

Lawson, E. C., Wadham, J. L., Lis, G. P., Tranter, M., Pickard, A. E., Stibal, M., Dewsbury, P. and Fitzsimons, S.: Identification and analysis of low molecular weight dissolved organic carbon in subglacial basal ice ecosystems by ion chromatography, Biogeosciences Discuss., 12(16), 14139–14174, 2015.

Montross, S., Skidmore, M., Christner, B., Samyn, D., Tison, J. L., Lorrain, R., Doyle, S. and Fitzsimons, S.: Debris-Rich Basal Ice as a Microbial Habitat, Taylor Glacier, Antarctica, Geomicrobiol. J., 31(1), 76–81, 2014.

Tison, J.-L. and Hubbard, B.: Ice crystallographic evolution at a temperate glacier: Glacier de Tsanfleuron, Switzerland, Geol. Soc. London, Spec. Publ., 176(1), 23–38, 2000.

Wainwright, J., Parsons, A. J., Cooper, J. R., Gao, P., Gillies, J. A., Mao, L., Orford, J. D. and Knight, P. G.: The concept of transport capacity in geomorphology, Rev. Geophys., 53(4), 1155–1202, 2015.

Yde, J. C., Finster, K. W., Raiswell, R., Steffensen, J. P., Heinemeier, J., Olsen, J., Gunnlaugsson, H. P. and Nielsen, O. B.: Basal ice microbiology at the margin of the Greenland ice sheet, Ann. Glaciol., 51(56), 71–79, 2010.